# The Prodigious Potential of mRNA Electrotransfer as a Substitute to Conventional DNA-Based Transient Transfection

**DOI:** 10.3390/cells12121591

**Published:** 2023-06-08

**Authors:** Théo Juncker, Bruno Chatton, Mariel Donzeau

**Affiliations:** UMR7242 Biotechnologie et Signalisation Cellulaire, Université de Strasbourg, F-67412 Illkirch, France

**Keywords:** mRNAs, electroporation, transfection efficiency, nanobody, transient transfection, cell imaging

## Abstract

Transient transfection of foreign DNA is the most widely used laboratory technique to study gene function and product. However, the transfection efficiency depends on many parameters, including DNA quantity and quality, transfection methods and target cell lines. Here, we describe the considerable advantage of mRNA electroporation compared to conventional DNA-based systems. Indeed, our methodology offers extremely high transfection efficiency up to 98% regardless of the cell line tested. Protein expression takes place a few hours post-transfection and lasts over 72 h, but overall, the electrotransfer of mRNAs enables the monitoring of the level of protein expressed by simply modulating the amount of mRNAs used. As a result, we successfully conducted cell imaging by matching the levels of expressed V_H_Hs and the antigen present in the cell, preventing the necessity to remove the excess unbound V_H_Hs. Altogether, our results demonstrate that mRNA electrotransfer could easily supplant the conventional DNA-based transient expression system.

## 1. Introduction

For many decades, foreign DNA transient transfection has been widely used to study gene products and functions in eukaryotic cells [1]. The most common routinely used methods for DNA transfection are cationic lipid and cationic polymer nano-carriers, which are up-taken by cells via endocytosis [2,3,4]. Besides the chemical methods for DNA delivery, several permeabilized-based membrane disruption delivery methods exist including electroporation [5]. Intracellular delivery by electroporation is based on the transient loss of the semi-permeability of cell membranes when subjected to electrical impulses [5,6]. Nevertheless, the transfection efficiency depends mainly on DNA quantity and quality [1]. The quality of DNA depends, among other factors, on the presence of endotoxins or lipopolysaccharides (LPS), which are byproducts of plasmid DNA preparation [7]. The presence of LPS is unlikely to significantly affect the transfection efficiency or cell proliferation of commonly used cell lines, but it may pose challenges for specific cell lines [7,8,9]. Nonetheless, the two major drawbacks of DNA-based transient transfection are the inability to carefully control the expression level of the foreign construct even by using weak versus strong promoters [10] and, for most cell lines, the inability to reach up to 95% of transfected cells [11]. An efficient alternative to the use of traditional DNA-based constructs for transient gene expression in eukaryotic cells is the exogenous delivery of messenger RNAs (mRNAs). The mRNA delivery has a high potential in many fields and has been previously reported for several cell lines, such as fibroblasts, dendritic cells, macrophages and various stem cells [12,13,14,15].

Here, we describe a highly efficient electrotransfection delivery system of in vitro-transcribed (IVT) mRNA to numerous cancer and immortalized cell lines. To challenge our methodology, we selected several biological models.

We transiently expressed different protein constructs localized either in the cytosol or in the nucleus. Apart from the classical fluorescent proteins eGFP and mScarlet, we examined the oncogenic protein E6 encoded by the human papillomavirus 16 (HPV16), mainly responsible for HPV carcinogenesis through degradation of the tumor suppressor p53 [16] and the high affinity con1 peptide (for consensus motif 1) that binds to the proliferating cell nuclear antigen (PCNA), which is essential for DNA replication and DNA repair [17,18]. We also evaluated a toxin fragment derivate from the exotoxin A from *Pseudomonas aeruginosa* [19,20]. Finally, we tested two well-characterized nanobodies or V_H_Hs (variable domain of heavy-chain only antibody), an anti-eGFP and an anti-Lamin [21]. Our results demonstrate a high potential of mRNA electroporation in living cells over DNA-based transient transfection strategy.

## 2. Results

### 2.1. High Transfection Efficiency for Various Cell Lines

We first evaluated two methods for the transient transfection of mRNA, the cationic polymers JetMESSENGER^®^ and the electroporation, and one method for DNA, the electroporation. Several cell lines were tested: two breast cancer cell lines, MDA-MB231 and HCC1954; the immortalized MRC-5 cell line; two HPV16 positive cell lines, Caski and SiHa; and two cell lines identified as very difficult to transfect with DNA, the natural killer NK-92 and the human gastric adenocarcinoma AGS cell lines [22,23]. The mScarlet fused to the SV40 nuclear localization signal (NLS) that enabled nuclear transport was used as a fluorescent marker. The cells were transfected with 500 ng of mRNA coding for the mScarlet-NLS using either JetMESSENGER^®^ or electroporation, or with 1 µg and 500 ng of plasmid DNA encoding the mScarlet using electroporation. The cells were fixed for 24 h after transfection and analyzed either with direct immunofluorescence or with flow cytometry. While approximately 100% transfection efficiency was achieved with mRNA electroporation for all tested cell lines, the transfection efficiency with the JetMESSENGER^®^ was particularly cell line dependent (Figure 1 and Appendix A). The efficiency was approximately 80% for the SiHa cell line but between 30% and 5% for the Caski and MDA-MB231 cell lines, respectively. We then evaluated the harmfulness of the electroporation versus the JetMESSENGER^®^ on cell viability. No significant difference could be observed, as both methods led to a percentage of survival of approximately 85% (Appendix A).

Surprisingly, DNA transfection efficiency was extremely low for the cell lines tested (Appendix A). With 1 µg, the transfection efficiency reached a maximum of 15% and dropped to 1.5% with 500 ng. In addition, it was very damaging to the cells, even though the DNA was endotoxin free. The percentage of survival was very low compared to mRNA electroporation, approximately 30 to 50% depending on the cell line (Appendix A).

To further validate the combination of electroporation and the advantages of mRNA transient transfection over DNA-based transient transfection, we evaluated, as a proof of concept, several proteins expressed in different cellular compartments.

### 2.2. E6 Oncoprotein from HPV16

We first examined the oncoprotein E6 encoded by the virus HPV16 (16E6). HPV carcinogenesis is mainly connected to the E6 and E7 oncoproteins, which interact with several host proteins [16]. In particular, HPV E6 hijacks the cellular ubiquitin ligase E6AP together with the tumor suppressor p53, which leads to ubiquitin-mediated degradation of p53 [24,25] (Figure 2A). Thus, we evaluated the levels of p53 with immunofluorescence after the electroporation of the mRNA coding for 16E6 in MDA-MB231 cells. MDA-MB231, a breast cancer cell line, was selected due to its notable expression level of the p53 protein. As shown in Figure 2B, transfection of the mRNA encoding the 16E6 protein led to a strong reduction in the p53 level in more than 99% of the cells, whereas the mRNA encoding the mScarlet had no effect (control). The expression of 16E6 was tested with western blot (Figure 2C) in the presence or absence of MG132. As expected, the 16E6 protein was barely detectable in the absence of the proteasome inhibitor. The expression of mScarlet was also tested with western blot (Figure 2C).

### 2.3. TOX Fragment from Pseudomonase aeruginosa

We next evaluated the cytotoxicity of a toxin fragment derivate from the exotoxin A from *Pseudomonas aeruginosa* that is used to engineer immunotoxins [19,20]. The toxin fragment (TOX) inactivates the eukaryotic translation elongation factor 2 (eEF-2) by ADP-ribosylation, inhibiting protein translation and thus triggering cell death [19]. A total of 100 ng of mRNA coding for TOX, comprising the active enzymatic domain from 400 to 613 aa or for an inactive form, ∆TOX (489 to 613 aa), deleted for the first 89 amino acids, was electrotransferred into HCC1954 and HeLa cells. HCC1954, a breast cancer cell line, was chosen because it had previously been used to evaluate the killing efficacy of the TOX module as a protein [1]. HeLa was chosen since it is a well-established and commonly used cell line in various research studies. Survival was measured 72 h following transfection using the MTT assay. As shown in Figure 3A, transfection of 100 ng of mRNA coding for TOX was sufficient to trigger more than 95% cell death. The amount of TOX mRNA was then titrated, starting with 40 ng down to 0.004 ng. As shown in Figure 3B, the toxin effect correlated with the amount of mRNA transfected with an IC50 of 0.2 (±0.03) ng of mRNA.

### 2.4. Nanobody Anti-GFP

Subsequently, we tested the feasibility of transfecting mRNA that encodes for a nanobody with the aim of directly visualizing the targeted antigen with immunofluorescence. One of the evaluated nanobodies was the well-characterized nanobody anti-GFP (herein after referred to as nano-GFP), which has been extensively used to trace proteins fused to eGFP in a cellular context [21]. However, a high level of expression generates an excess of nanobody compared to the targeted antigen, therefore, producing a strong unspecific signal. Therefore, we titrated the amount of mRNA encoding the nano-GFP-NLS-mScarlet by transfecting decreasing amounts of the mRNA into HeLa H2B2-GFP cells to determine the optimal amount of mRNA for an optimal detection of the H2B2-GFP antigen (Figure 4A). We started to detect a weak specific signal at 50 ng of mRNA and reached a clear strong signal at 100 ng (Figure 4A).

The amount of 100 ng was optimal to visualize the distribution of H2B2 during the different steps of the cell cycle (Figure 4B) with minimal background in the cytoplasm, indicating that the majority of the anti-GFP nanobodies successfully bound to the eGFP. Thus, the expression level of the nano-GFP-NLS-mScarlet could be carefully controlled by the mRNA amount transfected and could be matched to the antigen level inside the cells.

### 2.5. Interplay between PCNA, Con1 and DNA Polymerase η

We have previously evaluated ligands targeting the proliferating cell nuclear antigen (PCNA) [10]. PCNA is an ubiquitous eukaryotic protein essential for DNA replication and DNA repair, and its inhibition is considered to be a promising anti-cancer strategy [17,18]. PCNA assembles into a ring-shaped, homo-trimer that encircles the DNA and facilitates DNA processing by recruiting various factors mainly via association at its inter domain connector loop (IDCL), such as the DNA polymerase η (pol η) [26,27,28,29]. The DNA polymerase η is implicated in translesion DNA synthesis (TLS) and co-localizes with PCNA at replication foci after DNA damage [26]. A high affinity con1 peptide has also been designed to bind to the IDCL of PCNA [30]. We evaluated whether the interaction between PCNA and con1 could inhibit the recruitment of the DNA pol η to PCNA as reported previously [26,27,28,29]. The mRNA coding either for con1-NLS-mScarlet or for nano-GFP-NLS-mScarlet [21,31] was transfected at different concentrations in the MRC5 cell line constitutively expressing the DNA pol η fused to the enhanced yellow fluorescent protein (EYFP). We had previously shown that an excess of con1 expression compared to PCNA-targeted antigen causes a strong unspecific signal [10]. Therefore, we transfected a decreasing amount of mRNA that encodes for con1-NLS-mScarlet to determine the optimal mRNA concentration to be used. The optimal concentration for con1-NLS-mScarlet was found to be between 50 and 25 ng, much lower than for the nano-GFP fused to the NLS-mScarlet (Figure 5A). At 24 h following transfection, the cells were treated with UV-C and further incubated 8 h before fixation. Exposure to UV-C radiation causes DNA double-strand breaks and triggers the activation of DNA repair mechanisms, implicating DNA pol η [26]. As shown in Figure 2B, EYFP-pol η (green signal) was detected by the nano-eGFP-NLS-mScarlet V_H_H that binds with high affinity to EYFP [10] (red signal) at the replication foci visualized by the strong co-localized yellow speckles (Figure 5B). However, when the cells were transfected with the con1-NLS-mScarlet mRNA, DNA pol η was no longer recruited to the foci (Figure 5B) as expected [27,29]. In conclusion, controlling the amount of transfected mRNA enables the detection of PCNA replication foci without pre-treatment with CSK (cytoskeleton buffer) used to wash away unbound proteins prior to fixation (Appendix A). These results demonstrate that the amount of mRNAs transfected can be easily monitored to match the antigen concentration inside the cells.

We next transfected 50 ng of mRNAs encoding for the nano-Lamin fused to mScarlet in the MDA-MB231 cells to visualize the lamin A with direct immunofluorescence. The cells were incubated 24 h prior to fixation. The nano-Lamin-mScarlet gave a sharp labeling signal of the Lamin A and surrounded the nuclear compartment, and it was possible to follow the lamin A distribution during the different steps of the cell cycle (Figure 6).

We assessed the expression level of the nano-Lamin at different time points and with different amounts of mRNA transfected as indicated (Figure 7A). The nano-Lamin gave a strong labeling of lamin A in almost all cells at an mRNA concentration of 200 ng. The visualization of the lamin A could be detected as early as 6 h post-transfection at an mRNA concentration of 200 ng and then peaked at 24 h for a concentration between 200 ng and 50 ng. The expression was maintained until 48 h post-transfection and then gradually decreased. The appropriate amount of mRNA required to detect the target antigen with a minimal background depends on the duration of the incubation period. In particular, while 200 ng of mRNA proved to be optimal after 6 and 12 h, it was too high at 24 and 48 h.

We investigated the feasibility of expressing two distinct proteins within cells through the co-electroporation of their respective mRNAs. To that end, the mRNA encoding for con1 fused to eGFP was transfected together with the mRNA encoding for the nano-Lamin-mScarlet. The cells were treated with UV-C 24 h post-transfection and fixed 8 h later. As shown in Figure 7B, both mRNAs were expressed in the same cell. In the particular case of con1-eGFP, foci were clearly detected after UV-C treatment. This showed that the cells were able to express both proteins efficiently. Finally, quantification was performed with flow cytometry at 24 h post-transfection for an optimal mRNA concentration of 100 ng. As shown in Figure 7C, the single or double transfection rate was close to 95%, representing a very high transfection efficiency.

## 3. Discussion

Transient transfection of foreign DNA is the most widely used laboratory technique to study gene function and gene product in cells [1]. However, there are many factors influencing the transfection efficiency. One factor remains the inherent variability among cell lines [11]. In addition, DNA should be of high quality, purity, integrity and concentration [7,8]. For example, DNA preparation can be contaminated with endotoxins, which can alter the growth of some cell lines [8,9]. Lastly, the transfection method plays a critical role not only on the efficiency parameter but also on the cell survival. It has also been demonstrated that many of the reagents used for transfection are toxic for most cells [1,11]. To summarize, the transfection efficiency of DNA will depend on the DNA quantity and quality, on the transfection method and on the target cell line.

We established a protocol using electrotransfer of mRNAs and demonstrated the huge potential of the transient transfection of mRNA over conventional DNA-based systems by challenging our protocol with JetMESSENGER^®^ and with DNA electrotransfection. As a proof of concept, we transiently expressed six different constructs in various cell lines. For all constructs tested, we achieved extremely high transfection efficiencies of up to 99.0% in several cancer cell lines, including MDA-MB231, HCC1954, HeLa, Caski, SiHa and U-2OS, an immortalized MRC5 cell line and the immune cell line NK-92. The JetMESSENGER^®^ gave lower transfection efficiency while maintaining cell integrity. Surprisingly, DNA electroporation resulted in a dramatic/drastic decrease in transfection efficiency and cell viability despite the fact that the DNA was free of endotoxin.

With our method, the expression of the protein encoded by the mRNA occurs as early as 6 h after transfection and lasts up to 72 h. Indeed, the electroporation method allows the delivery of the mRNA inside the cytosol where it is rapidly translated. As the mRNAs are synthetized in vitro, no byproducts such as endotoxins are present. In addition to the very high transfection efficiency, we could monitor the protein expression level by carefully titrating the amount of transfected mRNA. Controlling the level of protein expression has a great potential for various applications including cell live imaging. Conventional methods of transfecting expression plasmids often result in an excess of expressed proteins compared to the intracellular targets leading to a very high non-specific signal. For example, the use of mRNAs enables the regulation of the quantity of nanobodies expressed in the cells, which in turn can match the level of the antigens without the need to remove excess unbound V_H_Hs. Indeed, we were able to precisely determine the amount of mRNA required for transfection, taking into account both the target antigen and the duration of the incubation period. A major drawback that may explain the limited focus on the use of mRNA for transient cell transfection is the requirement for meticulous operating procedures to prevent its degradation during synthesis and handling.

However, this methodology offers extremely high transfection efficiency and rapid expression a few hours post-transfection with the ability to monitor the expression levels of the different constructs by titrating the amount of transfected mRNA. Given these advantages, we believe that the RNA-based expression system has the potential to replace the conventional DNA-based transient expression system.

## 4. Materials and Methods

**Cell lines.** HCC1954 (ATCC CRL-2338) and MDA-MB-231 (ATCC HTB-26) cell lines were maintained as monolayers in RPMI without HEPES (Gibco) supplemented with 10% heat-inactivated fetal bovine serum (FBS) (Gibco) and Pen–Strep (Gibco). CaSki (ATCC CRL-1550), SiHa (ATCC HTB-35), HeLa-H2B2eGFP (ref), MRC5-EYFP-pol η (ATCC CCL-171) (ref), AGS (ATCC CRL-1739) and U-2OS (ATCC HTB-96) were maintained as monolayers in Dulbecco’s modified Eagle’s medium (DMEM) (1 g/L glucose) (Gibco, Billings, MT, USA) supplemented with 10% heat-inactivated FBS and Pen–Strep. NK-92 cell line (ATCC CRL-2407^TM^) was maintained in suspension in DMEM medium (1 g/L glucose) (Gibco) supplemented with 10% heat-inactivated FBS, Pen–Strep and interleukin-2 (IL-2). Cells were cultivated at 37 °C with 5% CO_2_ in a humidified atmosphere. All cells were mycoplasma free.

**Plasmids.** The sequences of the nanobody anti-GFP, the nanobody anti-lamin, the mScarlet and the con1 have been described previously [10,21,32]. The nano-lamin sequence was cloned in pETOM into NcoI-SpeI giving rise to pETOM-nano-Lamin. The mScarlet was amplified from the pET-mScarlet-E3 described elsewhere [32] with either the forward primers mScarlet-SpeI-For GGGCCCACTAGTGTGAGCAAGGGCGAGGCAGTGATC or NLS-GFP-SpeI-For GGGCCCACTAGTCCCAAGAAGAAGAGAAAGGTGGTGAGCAAGGGCGAGGAGCTG and the reverse primer mScarelt-NheI-Rev GTTACCGCTAGCCTTGTACAGCTCG. The PCR fragments were purified and cloned into SpeI-NheI of the pETOM-con1, the pETOM-nano-GFP-E3 and the pETOM-nano-Lamin-E3 giving rise to the pETOM-con1-NLS-mScarlet, the pETOM-nano-GFP-NLS-mScarlet and the pETOM-nano-Lamin-mScarlet. The pβ-mScarlet was constructed by introducing into the NcoI-SpeI restriction sites of the pβ-con1-eGFP [10] the mScarlet coding sequence.

**Plasmid preparation.** Plasmids for DNA electroporation were prepared using EndoFree Plasmid Maxi Kit (Qiagen, Hilden, Germany) following the manufacturer protocol. Endotoxin detection was performed using the ToxinSensor^TM^ Endotoxin Detection System (Genscript, Piscataway, NJ, USA).

**mRNA synthesis.** The mRNAs were transcribed from the T7 promoter of either a linearized pETOM vector with EcoR-I restriction enzyme containing the gene of interest or from a PCR fragment amplified using a forward primer containing the T7 promoter and an adequate reverse primer. Linearized plasmids or PCR fragments were purified using NucleoSpin Gel and PCR Clean-up (Macherey Nagel, Düren, Germany) according to the manufacturer protocol and used as a template for in vitro transcription (IVT). Briefly, reactions (20 µL) were prepared by mixing 1.5 µg of linearized plasmid or 0.5 µg of PCR fragment, ARCA/dNTP Mix to perform the capping reaction (ARCA analog to GTP ratio of 4:1) and T7 DNA polymerase (Biolabs, New England, Ipswich, MA, USA). Reaction mix was incubated 30 min at 37 °C. The DNA template was then digested by adding 2 µL of DNase I and incubated at 37 °C for an additional 15 min. The poly (A) tailing reaction was performed by adding to the 20 µL reaction the *E. coli* poly (A) polymerase and the poly (A) reaction buffer in a final volume of 50 µL (Biolabs, New England). The reaction mix was further incubated at 37 °C for 30 min. Finally, 25 µL of cold LiCl was added to the 50µL mRNA tailing reaction and incubated for 30 min at −20 °C. mRNAs were precipitated with centrifugation at 4 °C for 30 min at 16,000 RCF. After washing the pellet with 70% ethanol, mRNAs were resuspended in pure water and used for the transfection experiments. Quantification and quality control of the mRNAs were performed using Agilent RNA 6000 Nano Kit (Agilent, Santa Clara, CA, USA).

**Transfection of mRNAs and DNA.** The transfection experiments with mRNAs were conducted either with JetMESSENGER^®^ according to the manufacturer protocol (Polyplus, Illkirch-Graffenstaden, France) or with the Neon^TM^ transfection system (Thermofisher Scientific, Waltham, MA, USA). For electrotransfer, 1.10^5^ cells in a final volume of 10 µL of R buffer (Thermofisher Scientific) were mixed with 1 µL of mRNAs or DNA (at various concentration) and drawn up in the 10 µL Neon^TM^ pipette tip. The following electroporation parameters were applied: 1200 V, 20 ms and 1 rectangular pulse. The cells were transferred into 24-well plates containing glass coverslips. Where indicated, the cells were irradiated with 30 J/m2 UV-C 8 h before harvesting.

**Immunofluorescence.** The cells grown on coverslips were fixed with 4% (*w*/*v*) paraformaldehyde for 30 min. For cells observed with direct immunofluorescence, coverslips were mounted with Fluoromount G containing 4′,6′-diamidino-2 phenyleindole (SouthernBiotech, Birmingham, UK). For indirect fluorescence, cells were prior permeabilized with 0.2% Triton X-100 for 5 min and incubated with anti-p53 (7F5) rabbit mAb (Cell signaling) and a secondary anti-rabbit- Alexa 488 (Life Technology, Carlsbad, CA, USA). The treated cells were analyzed using conventional fluorescence microscopy with a Leica DM5500 microscope (Leica Microsystems, Wetzlar, Germany). Images were processed with Image J2.0.0.

**Fluorescence-Activated Cell Sorting (FACS).** NK-92 cells were washed twice in PBSx1 and analyzed on an AccuriTM C6 Plus flow cytometer (BD Bioscience, San Jose, CA, USA).

**Cytotoxicity Assay.** Cells transfected with mRNA were maintained 72 h in culture. A MTT assay was conducted according to the manufacturer’s protocol. The absorbance was measured in a Tecan reader (595 nm). IC50 values were calculated by fitting a sigmoidal model with R software. Wells with cells treated with PBS were set to 100% of cell viability.

**Statistical analysis.** The data are presented as the mean ± standard deviation (SD) of at least three independent replicates for each experiment. The statistical analysis was performed using GraphPad Prism 5.0 (GraphPad Software, La Jolla, CA, USA), applying one-way analysis of variance (ANOVA) and the Student’s *t*-test to compare the differences among the experimental groups.

## Figures and Tables

**Figure 1 cells-12-01591-f001:**
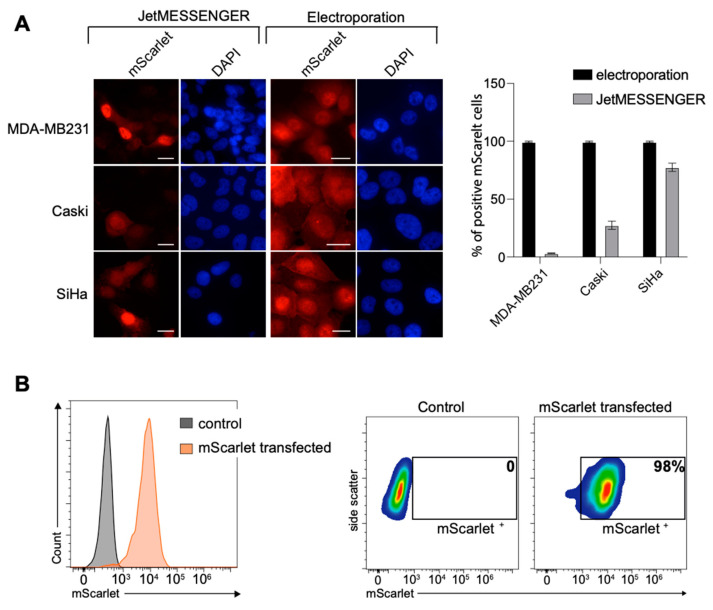
mRNA transient transfection efficiency. (**A**) Direct fluorescent detection of mScarlet at 24 h post-transfection. MDA-MB213, Caski and SiHa cells were transfected with mRNA coding for the mScarlet using JetMESSENGER^®^ or electroporation. Nuclei were stained with DAPI (blue). Scale bar, 20 μm. The percentage of transfection efficacy was quantified with ImageJ and blotted (n ≥ 60). (**B**) NK-92 cells were harvested 48 h post-transfection and analyzed with flow cytometry.

**Figure 2 cells-12-01591-f002:**
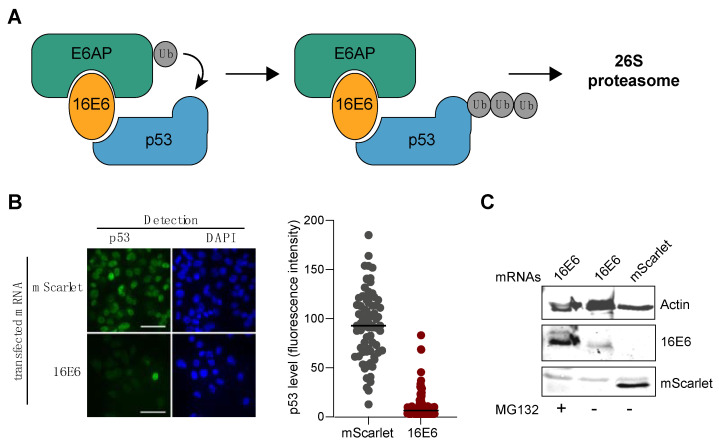
mRNA transient transfection efficiency. (**A**) Cartoon representing the degradation pathway of p53 via 16E6. (**B**) 24 h post-transfection of the 16E6 oncogenic protein or the mScarlet mRNAs; degradation of p53 was monitored with immunofluorescence using an anti-p53 rabbit (7F5) Ab followed by a secondary anti-rabbit Alexa448. Scale bar, 40 µm. Immunofluorescence images were quantified using Prism software (n ≥ 200). (**C**) Detection of 16E6 and mScarlet with western blot using 4C6 and penta-His primary Ab, respectively.

**Figure 3 cells-12-01591-f003:**
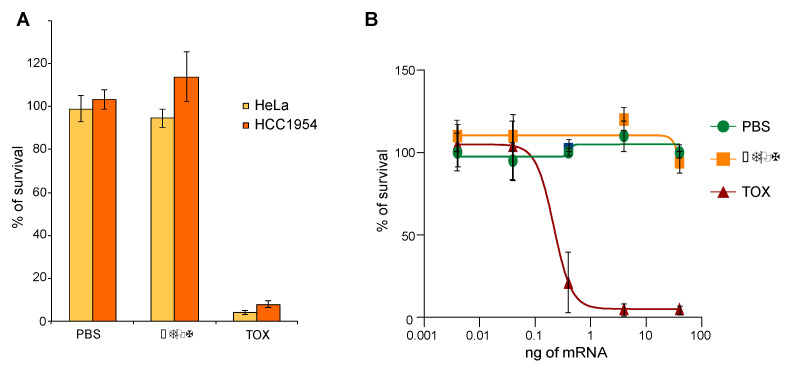
mRNA encoding for TOX delivered into cells. (**A**) 100 ng of mRNA coding for either ∆TOX or TOX was transfected in HeLa and HCC1954 cell lines. (**B**) Increasing amounts of mRNAs were transfected in HeLa cells. (**A**,**B**) Survival was measured 72 h post transfection using the MTT assay.

**Figure 4 cells-12-01591-f004:**
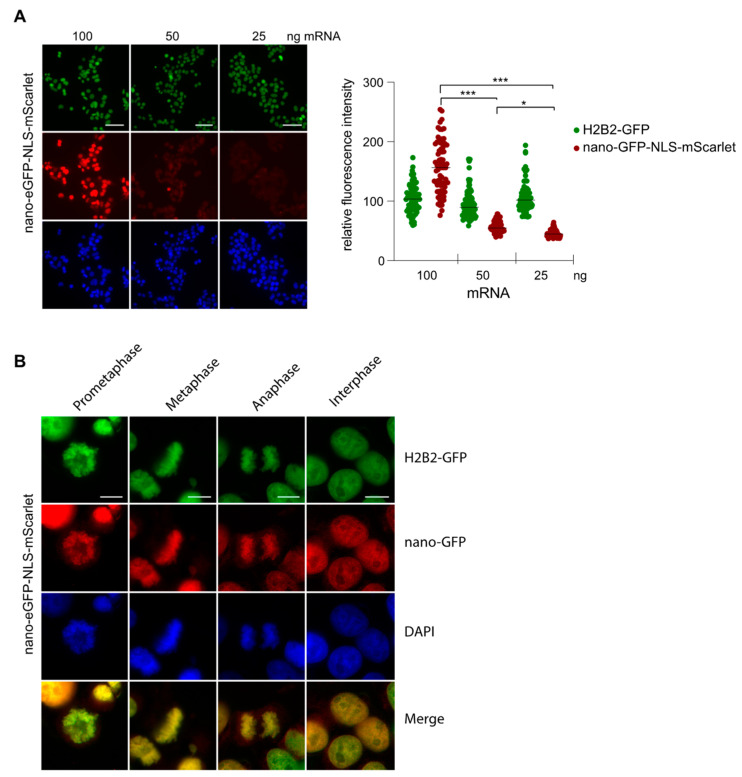
Visualization of the H2B2-GFP with the nano-GFP-NLS-mScarlet (**A**) The amount of mRNAs coding for nano-GFP fused to NLS-mScarlet transfected in the MDA-MB231 cells was titrated as indicated. Cells were fixed after 24 h and analyzed with fluorescence microscopy. Scale bar, 50 µm. The quantification was performed using ImageJ, and the statistical analysis was performed using GraphPad Prism 5.0. The Student’s *t*-test was performed: * *p* < 0.05, *** *p* < 0.001. (**B**) MDA-MB231 cells were transfected with 100 ng of mRNA encoding for the nano-GFP-NLS-mScarlet. After 24 h, cells were fixed and analyzed for the different phases of the cell cycle with direct fluorescence microscopy. Scale bar, 10 µm.

**Figure 5 cells-12-01591-f005:**
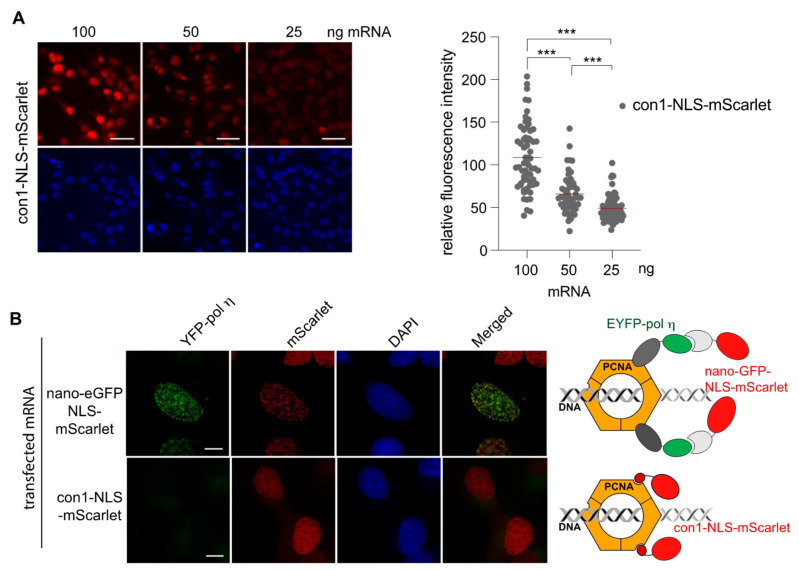
PCNA and DNA pol η detection. (**A**) Titration of the mRNAs coding for con1-NLS and nano-GFP-NLS fused to mScarlet. Cells were fixed after 24 h and analyzed with fluorescence microscopy. Scale bar, 40 µm. The quantification was performed using ImageJ, and the statistical analysis was performed using GraphPad Prism 5.0. The Student’s *t*-test was performed *** *p* < 0.001. (**B**) Cells were transfected with either the mRNA encoding nano-GFP-NLS-mScarlet or with con1-NLS-mScarlet as indicated. After 24 h, cells were treated with UV-C and fixed 8 h post-treatment. Foci were visualized with direct fluorescence. Cartoons are representing the binding of both partners to PCNA IDCL domain. Scale bar, 10 µm.

**Figure 6 cells-12-01591-f006:**
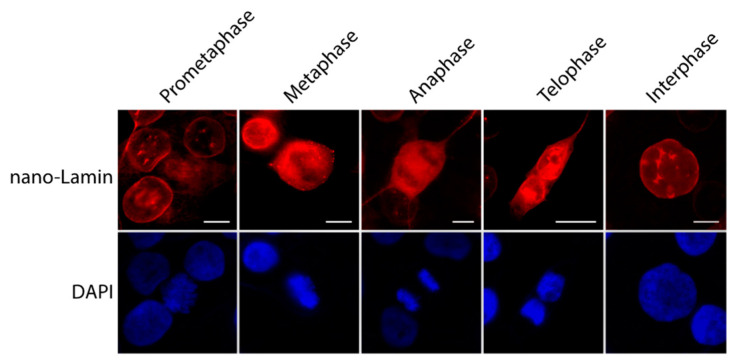
Cell cycle progression followed by nano-Lamin. MDA-MB231 cells were transfected with 50 ng of mRNAs coding for nano-Lamin-mScarlet. After 24 h, cells were fixed and analyzed with direct fluorescence microscopy. Scale bar, 10 µm.

**Figure 7 cells-12-01591-f007:**
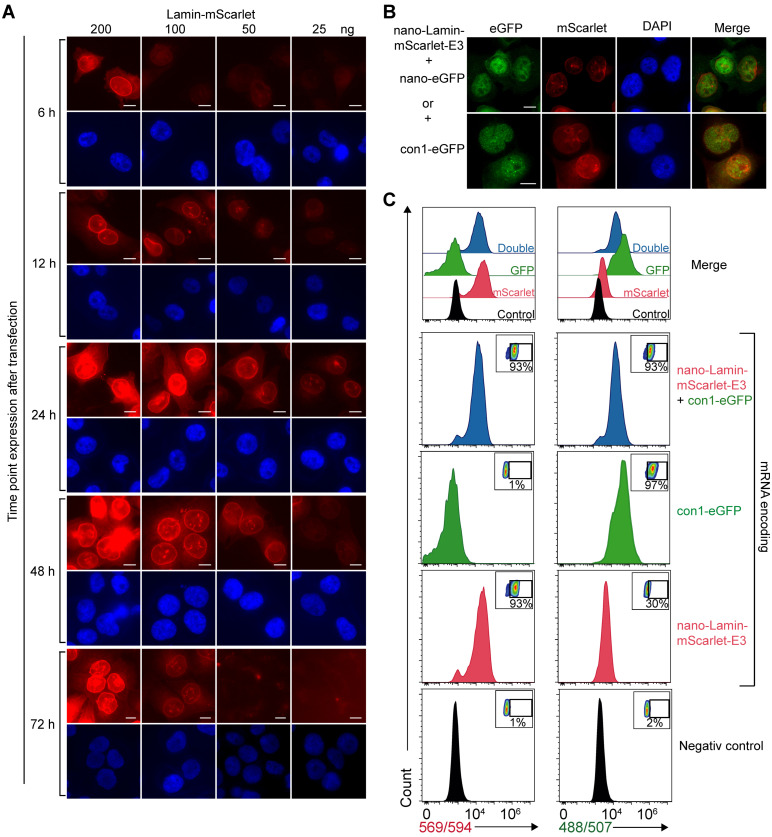
Efficacy of single and double transfection of mRNAs. (**A**) Titration of the mRNA encoding for nano-Lamin-mScarlet after different incubation time points. MDA-MB231 cells were transfected with different amounts of mRNAs as indicated. Cells were fixed after 6, 12, 24, 48 and 72 h of incubation. Images were captured with fluorescence microscopy, and quantification was performed with ImageJ. Scale bar, 10 µm. (**B**) Double detection of lamin A and either eGFP-NLS or con1. MDA-MB231 cells were transfected with 50 ng of mRNA coding for nano-Lamin-mScarlet-E3 together with either 100 ng of mRNA coding for eGFP-NLS or 50 ng of con1-NLS-eGFP. After 24 h, the cells were fixed and analyzed with direct fluorescence microscopy. Scale bar, 10 µm. (**C**) Quantification with flow cytometry of single transfection of 100 ng of mRNA encoding for either nano-Lamin-mScarlet or con1-eGFP or together as indicated. The percentages of transfected positive cells are noted in each FACS plot.

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
