# Peer review of "The Prodigious Potential of mRNA Electrotransfer as a Substitute to Conventional DNA-Based Transient Transfection"

_cells, 2023, doi:10.3390/cells12121591_

Round 1
Reviewer 1 Report
The manuscript is dealing with mRNA electrotransfer demonstrating high efficiency and the authors demonstrated this in many different cells and for many different applications.
The results are excellent, however it is difficult to evaluate them, as the authors provide very insufficient data about electroporation and most importantly, they did not compare the result to DNA electroporation. Namely, in the introduction, according to their statements, the man reason for evaluation of RNA electroporation, was due to the toxicity of DNA electroporation, due to the presence of endotoxin in the final plasmid preparation. I don’t agree with this statement, as the levels of endotoxins using appropriate kit for isolation is extremely low and it does not influence the growth of the cells. The authors should firmly support their statement with the references. In addition, in the introduction, they should introduce the electroporation, the method itself and its use for plasmid DNA and for RNA transfection.
This is a very premature paper.
Materials and methods section should contain more data about the cells – where were they obtained form, were they tested for mycoplasma. IN addition, what kind of mRNA they used, cap 1 or cap0?
The transfection system NeonR does not exist – it is only NeonTM
What kind of pulses did you used- rectangular, exponential? What kind of electrodes? Composition of the electrodes is important. What is the composition of R buffer?
Paragraph about statistical analysis is missing.
Why did you try only one electroporation protocol? What was the survival following electroporation of cells? The survival is shown for only two cell lines. I would recommend to show the survival for all cell Iines tested. In addition, the quantification was done only in Figure 1, for the rest application, the author stated that the quantification was performed, but they don’t show the data (for example fig 7 and fig 8).
The reason for choosing specific cell line for specific application should be described. For example, why was exotoxin tested in HeLa and HCC1954 cells?
The last paragraph in the introduction, should be in the discussion section.
Minor: the title of y axis is missing in Fig1B
Line 115: it’s ng, not pg.
Fig 3 – correct the legend
Discussion_ line 238 – you didn’t demonstrate a huge potential of mRNA over conventional DNA-based system, because you didn’t compare the two systems.
Explain the abbreviation VHHS.
Author Response
We would like to thank the reviewers for their helpful comments towards improving our manuscript. In the subsequent section, we highlight general concerns of reviewers and our effort to address these concerns. All modifications have been highlighted in blue in the text of the manuscript.
Reviewer 1
The results are excellent, however it is difficult to evaluate them, as the authors provide very insufficient data about electroporation and most importantly, they did not compare the result to DNA electroporation.
We have performed additional experiments to compare DNA and mRNA electroporation efficiency and survival, and have added the results in the main text and as supplementary figure S3.
Namely, in the introduction, according to their statements, the man reason for evaluation of RNA electroporation was due to the toxicity of DNA electroporation, due to the presence of endotoxin in the final plasmid preparation. I don’t agree with this statement, as the levels of endotoxins using appropriate kit for isolation is extremely low and it does not influence the growth of the cells. The authors should firmly support their statement with the references.
We have been more carefully with the statement about the presence of endotoxins and their effect on cell viability and transfection efficiency, and we also have added some references.
“The quality of DNA depends, among other factors, on the presence of endotoxins or lipopolysaccharides (LPS), which are byproducts of plasmid DNA preparation [7]. The presence of LPS is unlikely to significantly affect the transfection efficiency or cell proliferation of commonly used cell lines, but it may pose challenges for specific cell lines [7–9].»
In addition, in the introduction, they should introduce the electroporation, the method itself and its use for plasmid DNA and for RNA transfection.
Thank you for this suggestion; we have introduced the electroporation methodology in the introduction.
“Most common routinely used methods for DNA transfection are cationic lipid and cationic polymer nano-carriers, which are uptaken by cells via endocytosis.[2–4] Beside chemical methods for DNA delivery, several permeabilized-based membrane disruption delivery methods exist among which electroporation [5]. Intracellular delivery by electroporation is based on the transient loss of semi-permeability of cell membranes subjected to electrical impulses [5,6].”
Materials and methods section should contain more data about the cells – where were they obtained form, were they tested for mycoplasma. IN addition, what kind of mRNA they used, cap 1 or cap0?
In materials and methods, we have included the ATCC references of each cell-line used in this study. All cells were negatively tested for mycoplasma. We also described carefully the protocol used for capped mRNAs synthesis as requested by the reviewer.
The transfection system NeonR does not exist – it is only NeonTM
Thank you, we have corrected.
What kind of pulses did you used- rectangular, exponential? What kind of electrodes? Composition of the electrodes is important. What is the composition of R buffer?
We have completed as requested by the reviewer and described the devise used for the electroporation experiment. The type of electrode was neon tip; long pulse duration of 20ms, with a rectangular pulse.
It’s not possible to obtain the composition of the R buffer as all buffer compositions are proprietary from Thermofishcer scientic.
Paragraph about statistical analysis is missing.
We have added a paragraph in materials and methods about statistical analysis.
“The data are presented as the mean ± standard deviation (SD) of at least three independent replicates for each experiment. Statistical analysis was performed using GraphPad Prism 5.0 (GraphPad Software, La Jolla, CA), applying either one-way analysis of variance (ANOVA) or Student's t-test to compare differences among the experimental groups.”
Why did you try only one electroporation protocol? What was the survival following electroporation of cells? The survival is shown for only two cell lines. I would recommend to show the survival for all cell Iines tested.
We tested 3 different conditions; every condition, except the one described in this work, led to high transfection efficiency, but with an increase of cell death (data not shown). We have added survival for the NK92, the HCC1954, the U2OS and the MRC-5 cell lines in the supplementary Figure S2B.
In addition, the quantification was done only in Figure 1, for the rest application, the author stated that the quantification was performed, but they don’t show the data (for example fig 7 and fig 8).
As also requested by the reviewer 2, we performed quantification by flow cytometry for the experiment for the nano-Lamin-mScarlet-E3 (figure 7) and for the co-transfection experiment figure 8. These two figures were combined in a new figure 7.
The reason for choosing specific cell line for specific application should be described. For example, why was exotoxin tested in HeLa and HCC1954 cells?
Thank for the suggestion. We have added the reason for choosing specific cell line in each case, as mentioned in the main text in blue. For example:
« and two cell lines identified as to be very difficult to transfect with DNA, the natural killer NK-92 and the human gastric adenocarcinoma AGS cell lines [22,23].»
“MDA-MB231, a breast cancer cell line, was selected due to its notable expression level of the p53 protein »
« HCC1954, a breast cancer cell line was chosen because it had previously been used to evaluate the killing effecticacy of the TOX module as a protein [1]. HeLa was chosen since it is a well-established and commonly used cell line in various research studies.”
The last paragraph in the introduction, should be in the discussion ∠ section.
Thank you for the very good suggestion. We have added the last paragraph of the intro in the discussion.
Minor: the title of y axis is missing in Fig1B
Thank you, has been added.
Line 115: it’s ng, not pg.
Thank you, we have corrected as request.
Fig 3 – correct the legend
Thank you, has been corrected
Discussion_ line 238 – you didn’t demonstrate a huge potential of mRNA over conventional DNA-based system, because you didn’t compare the two systems.
We have now compared the two systems and added the required experiments requested (main text and figure S3.
Explain the abbreviation VHHS.
Thank you, has been explained in the introduction
Reviewer 2 Report
This study reports an mRNA electroporation strategy to achieve high transfection efficiency for various cell lines. Overall, the strategy presented here is interesting, but some improvements to the manuscript are still required before its acceptance.
Below are some specific comments.
1. In Figure 2, western blot data is still needed to confirm the expression of 16E6 oncogenic proteins and mScarlet.
2. In Figure 3, the graphic annotation for ∆TOX is unrecognizable.
3. In Figure 3, it’s good to show the expression of TOX and ∆TOX, and the ADP-ribosylation level of eEF-2 with western blot. Only a figure about cell survival is a bit weak.
4. In Figure 4A, the mScarlet signal below 100ng of mRNA decreases a lot, it’s not that “25ng or below gave rise to an unspecific signal”, so it’s better to rephrase the sentence in lines 130-132.
5. In line 203, “it was too high after 24 and 48 hours”, not “to”.
6. In lines 216-218, the authors claimed the transfection rate approaching 100%, it’s better to include the FACS data to support this.
7. In the Discussion part, it’s also good to discuss the drawbacks or shortcomings of the mRNA transfection strategy, especially the high standard operation procedures to avoid the degradation of mRNA.
Author Response
We would like to thank the reviewers for their helpful comments towards improving our manuscript. In the subsequent section, we highlight general concerns of reviewers and our effort to address these concerns. All modifications have been highlighted in blue in the text of the manuscript.
Reviewer 2:
- In Figure 2, western blot data is still needed to confirm the expression of 16E6 oncogenic proteins and mScarlet.
We have introduced a WB for the expression of 16E6 and mScarlet in figure 2 as requested. We also introduced a cartoon to explain the mechanism of p53 degradation via 16E6 as described in Martinez-Zapien, D et al. 2016. Thus, as shown in the cartoon, 16E6 associates with the ubiquitine ligase E6AP and p53, and is directly degraded via the proteasome pathway. It is why we have used MG132 to inhibit the proteasome.
- In Figure 3, the graphic annotation for ∆TOX is unrecognizable.
We have modified the graphic accordingly by changing the color of ∆TOX. And more a precise description of the construct has been added in the main text.
« 100 ng of mRNA coding for the TOX fragment comprising the active enzymatic domain from 400 to 613 aa, or for an inactive form, ∆TOX (489 to 613 aa), deleted for the first 89 amino-acids, was electro-transferred into HCC1954 and HeLa cells. »
- In Figure 3, it’s good to show the expression of TOX and ∆TOX, and the ADP-ribosylation level of eEF-2 with western blot. Only a figure about cell survival is a bit weak.
It is a very interesting point, however as pinpointed, the cells are dying very quickly and it is very difficult to detect the active toxin fragment as it has been previously described even with a very good specific mAb [11]. Moreover, we wanted to show that every cell expressed the protein of interest (nearly 100% transfection efficacy). The IF done with an anti-His6 to detect the toxin was not conclusive because of the high background of this mAb and on the cell state (dying).
- In Figure 4A, the mScarlet signal below 100ng of mRNA decreases a lot, it’s not that “ 25ng or below gave rise to an unspecific signal”, so it’s better to rephrase the sentence in lines 130-132.
We agree with the reviewer and rephrase the sentence « We started to detect a weak specific signal at 50 ng of mRNAs and reached a clear strong signal at 100 ng (Figure 4A).”
- In line 203, “it was too high after 24 and 48 hours”, not “to”.
Thank you, it has been corrected
- In lines 216-218, the authors claimed the transfection rate approaching 100%, it’s better to include the FACS data to support this.
We have included three examples of flow cytometry: one using nano-Lamin-mScarlet, one the con1-eGFP and lastly both constructs in a new Figure 7. The transfection rate is approaching 95%.
- In the Discussion part, it’s also good to discuss the drawbacks or shortcomings of the mRNA transfection strategy, especially the high standard operation procedures to avoid the degradation of mRNA.
We have added the following statement in the conclusion as requested.
“A major drawback that may explain the limited focus on the use of mRNA for transient cell transfection is the requirement for meticulous operating procedures to prevent its degradation during synthesis and handling.”
Round 2
Reviewer 1 Report
The authors adequately addressed all my concerns and comments and I recommend to accept the manuscript in current form for publication.
Reviewer 2 Report
the manuscript has improved a lot compared to the last version, all my comments are addressed, i don't have further questions